



# A study of Mie scattering modelling for mixed phase Polar Stratospheric Clouds

Francesco Cairo[1], Terry Deshler[2], Luca Di Liberto[1], Andrea Scoccione[3], and Marcel Snels[1]

[1]Istituto di Scienze dell'Atmosfera e del Clima, Consiglio Nazionale delle Ricerche, Rome, Italy
[2]Department of Atmospheric Science, University of Wyoming, Laramie,Wyoming, USA
[3]Centro Operativo per la Meteorologia, Aeronautica Militare, Pomezia, Italy

**Correspondence:** Francesco Cairo (f.cairo@isac.cnr.it)

**Abstract.** Mie scattering codes are used to study the optical properties of Polar Stratospheric Clouds (PSC). Backscattering and extinction can be computed once the particle size distribution (PSD) is known and a suitable refractive index is assumed. However, PSCs often appear as external mixtures of Supercooled Ternary Solution (STS) droplets, solid Nitric Acid Trihydrate (NAT) and possibly ice particles, making questionable the use of Mie theory with a single refractive index and with the under-

lying assumption of spherical scatterers. Here we consider a set of fifteen coincident measurements of PSC above McMurdo Station, Antarctica, by ground-based lidar and balloon-borne Optical Particle Counters (OPC), and in situ observations taken by a laser backscattersonde and an OPC during four balloon stratospheric flights from Kiruna, Sweden. This unique dataset of microphysical and optical observations allows to test the performances of Mie theory under fairly reasonable corrections when aspherical scatterers are present.

Here we consider particles as STS if their radius is below a certain threshold value $R_{th}$ and NAT or possibly ice if above it. The refractive indexes are assumed known from literature. Moreover, the Mie result for solid particles are reduced by a factor C <1, which takes into account the backscattering depression expected from the asphericity. Finally, we consider the fraction X of the backscattering from the aspherical part of the PSD as polarized, and the remaining (1-X) as depolarized. The three parameters $R_{th}$, C and X of our model are chosen to provide the best match with the observed optical backscattering

and depolarization. The comparison of the calculations with the measures is satisfactory for the backscattering but not for the depolarization, and possible causes are discussed. The results of this work help to understand the limits of the application of Mie theory in modeling the optical response of particles of different composition and morphology.

## 1   Introduction

Polar Stratospheric Clouds (PSC) appear in the polar stratosphere during winters due to the very low temperatures and the dynamic isolation of air within the polar stratospheric vortex. They have raised much attention due to the twofold role played in polar stratospheric ozone depletion: providing surfaces for the heterogeneous reactions that lead to the reactivation of chlorine



and decreasing the concentration of $HNO_3$ in the gaseous phase thus altering the balance of the chlorine activation/deactivation cycles (Solomon, 1988). A comprehensive review of studies and knowledge acquired can be found in Tritscher et al. (2021).

PSCs can either be formed of liquid droplets composed of supercooled ternary solutions (STS) of sulfuric acid, nitric acid and water, or solid nitric acid trihydrates (NAT), the thermodynamically stable form of $HNO_3$ and $H_2O$ in the polar stratosphere, or possibly - when temperature is low enough - ice. Initially PSCs were classified as three types based on lidar measurements of the intensity of the backscattered light and the amount of depolarization of the returned signals (Browell et al., 1990). With the accumulation of observations, it has been realized that it is not common to observe a PSC of a well defined type (Pitts

et al., 2018b). More often PSCs appear as external mixtures of liquid STS droplets, NAT, and/or ice, depending on the thermal history that led to their formation. For example, it is believed that the nucleation of NAT could start in droplets of a pre-existing population of STS, but not all liquid droplets may convert into solid NATs, allowing the coexistence of particles of different composition and phases and thus of intermediate optical characteristics (Peter and Grooß, 2012).

The existence of multi-phase PSC with particles of different shapes and composition, henceforth with different particle

refractive indices, makes the modeling of the scattering characteristics of the cloud problematic. While Mie scattering theory has been used to analyse PSCs consisting of spherical particles (Toon et al., 2000), detailed analysis of observations of different classes of PSCs (Deshler et al., 2000) may be questionable because they may consist of non spherical solid particles with sizes comparable to the wavelength of the laser, and hence the results may be hampered by biases due to the unverified assumption of spherical scatteres. Because of this, some studies have chosen to limit themselves to liquid clouds only (Jumelet et al.,

2008), or to make an effort and use theoretical modelling of light scattering for aspherical scatterers in the analysis. A viable solutions which is not so demanding in term of computational effor is the use of T-matrix theory (see Liu and Mishchenko (2001), and references within). T-matrix method is an exact technique for the computation of nonspherical scattering based on a direct solution of Maxwell's equations assuming homogeneous, rotationally symmetric non spherical particles or clusters of spheres (Mishchenko et al., 1996), and T-matrix codes are orders of magnitude faster than other approaches used in particle

light scattering, like the Discrete Dipole Approximation (Singham and Salzman, 1986) and the Finite Difference Time Domain (Yang and Liou, 1996) techniques. The T-matrix approach to compare microphysical and optical observations has been used in a number of cases (Voigt et al., 2003; Scarchilli et al., 2005), under the assumption of particles as prolate or oblate spheroids. However even this approach could be questionable given that, just as solid particles cannot be modeled as spheres, for similar reasons it is doubtful that they can be modeled as spheroids, and biases can arise under that assumption as well. For instance,

Reichardt et al. (2002) showed how under the hypothesis of spheroidal particle shapes, surface area density and volume density of leewave PSCs are systematically smaller by, respectively, 10–30% and 5–25% than the values found for mixtures of droplets, asymmetric polyhedra, and hexagons. Furthermore, there is no clear indication on what kind of aspect ratio can be univocally assumed for the spheroidal case (Reichardt et al., 2004; Engel et al., 2013; Woiwode et al., 2016).

Given that there is still no completely satisfactory solution to tackle scattering from solid PSC particles due to the ambigui-

ties that still persist about their shape, the use of Mie theory continues to be attractive and is used widely despite the unverified hypothesis of spherical diffusers. The speed of computation it offers is advantageous when used for statistical or climatolog-



ical studies (Pitts et al., 2009, 2013), or when optical properties, less dependent on the assumption of sphericity, have to be calculated, as is in the case of extinction (David et al., 2012; Daerden et al., 2007).

The aim of our study is to employ concomitant microphysical and optical measurements of PSCs to understand the capabilities and limits of the use of Mie theory when both liquid and solid particles are present. This in order to identify possible corrections to Mie results and to evaluate their performance. The methodology illustrated is not restricted to the study of mixed phase PSC, but can find applications in all those cases in which the aerosol appears as an external mixture of solid and liquid particles, distinguishable on the basis of their different typical sizes.

## 2 Data and Methods

### 2.1 Dataset

We have studied fifteen coincident measurements by groundbased lidar and balloon-borne Optical Particle Counters (OPC) taken above McMurdo Station, Antarctica between 1994 and 1999 (Snels et al., 2021) and four in situ balloonborne observations taken by laser backscattersonde and OPC during 4 balloon stratospheric flights from Kiruna, Sweden, between 2000 and 2002 (Weisser et al., 2006). The Antarctic lidar and balloonborne OPC dataset has been extensively discussed in Snels et al. (2021) where it has been used to provide empirical relations linking particle Surface Area and Volume densities with the backscatter coefficients. The main characteristics of the instrumentation will be here only briefly recalled.

The lidar observations have been provided by a system detecting 532 nm backscattered light with parallel and perpendicular polarization with respect to the linear polarization of the emitting laser (Di Donfrancesco et al., 2000), thus allowing the measurement of Backscatter Ratio BR, Volume Depolarization $\delta$ and Aerosol Depolarization $\delta_A$ from 10 to 23 km. These optical parameters follows the usual definitions (Cairo et al., 1999); in the following the subscripts $mol$ and $A$ denote respectively the molecular and particle contribution to the optical coefficients, and $cross$ and $par$ denote the perpendicular and parallel polarization of the Backscatter Coefficient $\beta$ (Collis and Russell, 1976).

$$BR = \frac{\beta_A^{cross} + \beta_{mol}^{cross} + \beta_A^{par} + \beta_{mol}^{par}}{\beta_{mol}^{cross} + \beta_{mol}^{par}} \tag{1}$$

$$\delta = \frac{\beta_{mol}^{cross} + \beta_A^{cross}}{\beta_{mol}^{par} + \beta_A^{par}} \tag{2}$$

$$\delta_A = \frac{\beta_A^{cross}}{\beta_A^{par}} \tag{3}$$

$$\tag{4}$$

An alternative definition of Total Volume Depolarization $\delta_T$ and Total Aerosol Depolarization $\delta_{TA}$ will also be used in the following and is here introduced as:

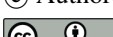



$$\delta_T = \frac{\beta_A^{cross} + \beta_{mol}^{cross}}{\beta_A^{cross} + \beta_{mol}^{cross} + \beta_A^{par} + \beta_{mol}^{par}} \tag{5}$$

$$\delta_{TA} = \frac{\beta_A^{cross}}{\beta_A^{par} + \beta_A^{cross}} \tag{6}$$

$$\tag{7}$$

being:

$$\delta_T = \frac{\delta}{\delta + 1} \tag{8}$$

The formulas for switching from one to the other can be found in Cairo et al. (1999).

The BR is retrieved using the Klett algorithm where the attenuation correction follows Gobbi (1995). The depolarization is calibrated with the method described in Snels et al. (2009). Uncertainties in BR are estimated to be 5%, but not less than 0.05, while uncertainty in $\delta_A$ is about 10-15% (Adriani et al., 2004). Typical measurements are 30–60 min integration over the signal, and the vertical resolution is 75 m in 1994 and 1995 and 225 m in the other years. For comparison with the OPC data, which are averaged to a vertical resolution of 250 m - which, given the balloon ascent rate, correspond approximately to a 60 second time average - the OPC data have been interpolated on the vertical grid of the lidar data.

The OPC is described in Hofmann and Deshler (1991) and Deshler et al. (2003a). A throughful revision of its dataset is presented in Deshler et al. (2019). The instrument uses white light to measure scattering at 40° in the forward direction from particles passing through a dark field microscope. Mie theory and a model of the OPC response function are used to determine particle size throughout the range from 0.15 to 10.0 $\mu$m radius. OPC provide time series of size resolved particle concentration histograms at 8 to 12 size bins, depending on the instrument used. A measurement of total concentration of particles is simultaneously determined by a CN counter close to the OPC, which grows all particles larger than 0.01 $\mu$m to an optically detectable size and counts them Campbell and Deshler (2014). Particle size histograms are fitted to unimodal or bimodal lognormal size distributions, which are the representation of size distribution used in this work.

A series of balloon launches were carried out from Kiruna, Sweden in the early 2000s. The payload included, among other instruments and in addition to the OPC and the CN, a backscattersonde capable of measuring in situ backscattering and depolarization at 532 nm with 10 seconds time resolution. Details of such instrument are presented in Adriani et al. (1999) and Buontempo et al. (2009). Here we use data from four flights that took place on 19 January 2000 (Voigt et al., 2003), 9 December 2001 (Deshler et al., 2003b), 4 and 6 December 2002 (Larsen et al., 2004; Weisser et al., 2006). As these balloon flight were not simple ascents like the antarctic ones, but were commanded to perform altitude changes by deflating the balloon and releasing ballast to maximize the transit time in the detected PSCs, for the purpose of the present work the backscattersonde data have been interpolated to the OPC 250 m average of the data, corrisponding to a time grid spacing of 60 or 75 seconds depending on flight.

In our study each data point includes the values of BR, $\delta_A$ and of the PSD defined by the three or six parameters of a mono or bimodal lognormal distributions. Altitude, pressure and temperature are also present as ancillary data. We identify a data



point as a PSC observation when the BR is greater than 1.2 and the temperature at the observation is below 195 K. Moreover, to select the presence of mixed phase clouds, we require that $\beta_A^{cross}$ is greater than $2\,10^{-6}km^{-1}sr^{-1}$. Under these condition, a total of 141 data point from the Antarctic and 332 data point from the Arctic flights have been selected for the study.

## 2.2 Optical model

While the nucleation of NAT and ice, is a threshold process, STS particles can grow upon cooling from the the ubiquitous
liquid Stratospheric Sulphate Aerosol (SSA), by continuously taking up nitric acid from the gas phase. They form droplets with volumes varying with temperatures but nevertheless with larger number density and smaller particle dimensions than NAT. Conversely, NAT particles are expected to be of smaller number density, but with dimension that can easily grow larger than the average STS particle radius of a few tenths of $\mu$m (Carslaw et al., 1997; Grooß et al., 2014) due to the smaller saturation vapour pressure of the nitric acid with respect to them. Deshler et al. (2003b) provide direct observations of this separation
between STS and NAT. All the more reason, due to the larger availability of water vapor, ice particles can grow even larger, often with linear dimensions exceeding 4–5 $\mu$m (Tritscher et al., 2019).

Our model tries to take advantage of the fact that in a mixed phase PSC the large particles are likely NAT or ICE, solid particles which depolarize the backscatter light, while the small ones are STS which are liquid, i.e. spherical, and do not depolarize the backscattering. Figure 1 shows our dataset mapped in terms of the BR - as 1-1/BR for the sake of clarity - and
$\delta_A$ and colour coded with respect to the ratio of the particles larger than 1 $\mu$m in radius, to the total number of particles. As can be seen, it is a general feature that at any one R value, the highest values of the depolarization occur when the fraction of large particles increase. The largest fraction of large particles at high R giving a depolarization which is lower than the one associated with smaller fraction at mid R suggests that $\beta_A^{cross}$ increases less than $\beta_A^{par}$ when the number of large particles increases, i.e. $\delta_A$ <100%. It has also to be noted that, in the high BR range, there are few cases where high depolarization is
associated with a low ratio of particles larger than 1 $\mu$m to total particle number. Inspecting these cases in greater detail, we note that although the relative abundance of particles larger than 1 $\mu$ was low, the presence of unusually very large particles is recorded, with dimension exceeding few $\mu$m. These very large particles, although in small concentrations, are believed to cause the high depolarization that is observed. The temperature in these few cases was however high enough to exclude ice particles.

In our model, particles are considered STS when their radius is below a threshold value $R_{th}$ and NAT above it. We use values of 1.44, and 1.48 for the refractive index of, respectively, STS, and NAT. These values are compatible with the large PSC data set produced by the CALIPSO observations (Hoyle et al., 2013; Pitts et al., 2018a) and fall within the estimates presented for STS and NAT (Adriani et al., 1995; Deshler et al., 2000; Scarchilli et al., 2005). For completeness, ice particles are considered for radii larger than 4 only when temperatures fell below 185 K. This happens only in 10% of the total dataset. In those few
cases a value of 1.31 was used (Kokhanovsky, 2004).

For each data point, we split the $PSD$ into two branches, namely $PSD_{STS} = PSD(r < R_{th})$ and $PSD_{asph} = PSD(r > R_{th})$. As stated, the presence of ice particles is taken into consideration by inspecting the temperature $T$ observed at the measurement, so if $T > T_{ICE}$ we pose $PSD_{NAT} = PSD_{asph}(r > R_{th})$, while if $T < T_{ICE}$ we limit the presence of NAT





particles at radii smaller than $4\mu$m, i.e. $PSD_{NAT} = PSD_{asph}(R_{th} < r < 4\mu$m$)$ and consider as ice the particles with bigger
radii, $PSD_{ICE} = PSD_{asph}(r > 4\mu$m$)$.

For the Mie backscattering of the portion of PSD assumed to be composed of solid particles, we multiply the result by a depression factor C <1 which takes into account the $\beta_A^{par}$ reduction expected from aspheric particles (Liu and Mishchenko, 2001; Mishchenko et al., 1996).

Finally we consider the fraction X of the backscattering contribution of the aspherical particles as polarized, i.e. retaining
the same polarization of the incident light, and the remaining (1-X) as depolarized.

The backscattering coefficients for the STS, NAT and ICE particles are separately computed with a Mie code (Bohren and Huffman, 2008) as: $\beta_{STS} = \beta^{Mie}(PSD_{STS}) = \beta_{STS}^{Mie}$, $\beta_{NAT} = C \cdot \beta^{Mie}(PSD_{NAT}) = C \cdot \beta_{NAT}^{Mie}$ and possibly $\beta_{ICE} = C \cdot \beta^{Mie}(PSD_{ICE}) = C \cdot \beta_{ICE}^{Mie}$, bearing in mind that, while $\beta_{STS}$ is fully polarized, the latter two can in fact be written as sums of a polarized and a unpolarized part

$$\beta_{NAT} = \beta_{NAT}^{par} + \beta_{NAT}^{cross} = C \cdot X\beta_{NAT}^{Mie} + C \cdot (1-X) \cdot \beta_{NAT}^{Mie} \tag{9}$$

$$\beta_{ICE} = \beta_{ICE}^{par} + \beta_{ICE}^{cross} = C \cdot X\beta_{ICE}^{Mie} + C \cdot (1-X) \cdot \beta_{ICE}^{Mie} \tag{10}$$

The total particle backscattering from the particles of the PSD can thus be written as:

$$\beta_A = \beta_{STS} + \beta_{NAT} + \beta_{ICE} \tag{11}$$

Here we have used for the sake of simplicity the same X and C for NAT and ICE. Even if this hypothesis were not fully
verified, this should not impact severely our study, as only 10% of our observations have temperatures below 185 K, and no ice observations could be clearly discerned in our database. Now, the polarized and depolarized components of the backscattering from the particles described by the total size distribution PSD can be written as:

$$\beta_{Apar} = \beta_{STS}^{Mie} + C \cdot X \cdot [\beta_{NAT}^{Mie} + \beta_{ICE}^{Mie}] \tag{12}$$

$$\beta_{Across} = C \cdot (1-X) \cdot [\beta_{NAT}^{Mie} + \beta_{ICE}^{Mie}] \tag{13}$$

and the total particle depolarization $\delta_{TA}$ is:

$$\delta_{TA} = \frac{C \cdot (1-X) \cdot [\beta_{NAT}^{Mie} + \beta_{ICE}^{Mie}]}{\beta_{STS}^{Mie} + C \cdot X \cdot [\beta_{NAT}^{Mie} + \beta_{ICE}^{Mie}] + C \cdot (1-X) \cdot [\beta_{NAT}^{Mie} + \beta_{ICE}^{Mie}]} \tag{14}$$

In this framework, in the case only aspherical particles are present, the particle depolarization attains the limiting value $\delta_{TA}^{asph}$

$$\delta_{TA}^{asph} = \frac{C \cdot (1-X) \cdot [\beta_{NAT}^{Mie} + \beta_{ICE}^{Mie}]}{C \cdot [\beta_{NAT}^{Mie} + \beta_{ICE}^{Mie}]} = 1 - X \tag{15}$$





which is independent of C. The knowledge of $\delta_{TA}^{asph}$ would allow us to determine X and hence reduce to two the free
parameters of our optical model.

## 2.3 Determination of the polarized fraction

The measurement of a PSC composed exclusively of solid particles is a rare and above all uncertain event, as we can never
be completely certain of the absence of liquid aerosols. However, Adachi et al. (2001) demonstrated that in a plot showing
the Total Volume Depolarization $\delta_T$ towards 1-1/BR, the experimental points of solid, liquid or variously mixed PSCs are
distributed within a triangle whose vertices are 0, $\delta_{TA}^{asph}$ and $\delta_{mol}$. These vertexes represent respectively the value of the
total volume depolarization in the case of pure liquid clouds and pure solid clouds for $BR = \infty$, when the total volume
depolarization coincides with the total aerosol depolarization, and in the case when no particles are present i.e. when the
total volume depolarization attains its molecular value $\delta_{mol}$ (Young, 1980). Hence the extrapolated intercept on the y axis at
$BR = \infty$ is precisely $\delta_{TA}^{asph}$. This procedure allows us to estimate its value. This requires the assumption that the experimental
points that fill the triangle of vertices defined above represent PSC observations in mixed phase in which all solid particles share
the same aerosol depolarization. Alternatively, one can interpret the presence of the datapoints filling the triangle as produced
by PSCs of solid particles with various different depolarizations. At present there is no way to prove the validity of one or the
other of the interpretations, or the reality of both.

Figure 2 reports total volume depolarization towards 1-1/BR from twelwe years of lidar observations from 1990 to 2002
in the antarctic station of McMurdo (Adriani et al., 2004). Despite the dispersion of the experimental points, a value close to
40% for $\delta_{TA}^{asph}$ can be tentatively assumed - the corrisponding value for $\delta_A$ being 67% according to (6) - which, following (15),
suggest to fix the value of the polarized fraction X to 0.6.

## 2.4 Variability with the threshold radius $R_{th}$ and aspherical depression factor C

After the determination of X, we have computed $\beta_A$ and $\delta_A$ for a set of threshold radius $R_{th}$ and aspherical depression factor
C ranging respectively from 0.25 to 2 $\mu$m and from 0 to 1. To find the values that provide the best match of the computed $\beta_A$
and $\delta_A$ with the measured ones $\beta_A^{meas}$ and $\delta_A^{meas}$, we have calculated the respective Mean Squared Errors (MSE), as:

$$\sigma_{\beta_A}^2 = \frac{\sum_{i=1}^n (\beta_{Ai} - \beta_{Ai}^{meas})^2}{n} \tag{16}$$

$$\sigma_{\delta_A}^2 = \frac{\sum_{i=1}^n (\delta_{Ai} - \delta_{Ai}^{meas})^2}{n} \tag{17}$$

Covariance has also been computed resulting to be negligible, except for $R_{th}$ less than 0.5 $\mu$m, where for C smaller than
0.05 resulted positive, and for C between 0.1 and 0.7, slightly negative (correlation between -0.3 and 0).

## 3 Results

Figure 3 and 4 reports the color coded MSEs, with respect to the threshold radius $R_{th}$ and the aspherical depression factor
C. Such plots allows to select the best choice of $R_{th}$ and C to match computations with observations. In fact, while the $\sigma_{\delta_A}^2$





basically only suggest to avoid the smallest range of $R_{th}$ and the largest C, $\sigma_\beta^2$ defines an optimal region for the two parameters,
namely $R_{th}$ between 0.4 an 0.6 $\mu$m and C between 0.2 and 0.5.

It is appropriate to ask how much these results depend on the choices made so far on the limits to the BR and $\beta_A^{cross}$ we have imposed on our dataset, and on the value of X previously obtained. We ran sensitivity tests, inspecting changes of the results by increasing X to 0.8 (which would correspond to $\delta_{TA}^{asph}$=20%, $\delta_A^{asph}$=25% ) and decreasing it to 0.5 (which would correspond to $\delta_{TA}^{asph}$=50%, $\delta_A^{asph}$=100%). We remind that these two cases represent respectively a decrease and an increase in the depolarized
fracton of the single scattering event. While the appearance of $\sigma_{\beta_A}^2$ is not affected by changes in X, as should be expected from the definition (11), not so $\sigma_{\delta_A}^2$, which for the large X tend to minimize more markedly at C greater than 0.8 and $R_{th}$ between 0.7 and 1 $\mu$m.

If we keep X fixed at 0.6 and reduce the experimental points admitted to the study, a choice that could be suggested by the desire to take non-marginal PSC events and with a more significative presence of both phases, by increasing the lower
acceptance limit for BR to 1.5 and for $\beta_A^{cross}$ to 5 $10^{-6}km^{-1}sr^{-1}$, we get no noticeable difference in the shape of $\sigma_{\delta_A}^2$ while the optimal region for the parameters that minimize $\sigma_{\beta_A}^2$ gets slightly shifted to larger $R_{th}$ between 0.5 an 0.7 $\mu$m and C between 0.3 and 0.5.

Optimal intervals can now be suggested for the choice of $R_{th}$ and C. Even if an unambiguous choice is not possible, it seems reasonable to assume the values 0.5 $\mu$m and 0.5. respectively, which is what we can also expect from what we know about
PSC microphysics (Voigt et al., 2000) and about the comparison between backscattering from spherical and scattering from aspherical vs spherical particles (Mishchenko, 2009). With this choice we can present in figures 5 and 6 the scatterplot $\beta_A$ and $\delta_A$ measured vs computed.

## 4   Discussion

Figure 5 reports the scatterplot of measured vs modelled $\beta_A$ and represent the analogue of figure 4 in Snels et al. (2021),
where in the present case we have used a larger dataset including Arctic balloon flights, and basically confirms the conclusions reported there. While the agreement between calculations and measurements can be considered satisfactory for $\beta_A$, it is certainly not so for $\delta_A$ as can be clearly seen from figure 6 which reports the scatterplot of its measured and modelled values. Although there is some tendency to fill the lower triangle of the figure, and therefore to have a linear relationship between the two parameters for at least some experimental points, the calculation often tends to overestimate the measured depolarization.
We can certainly invoke a noisiness in the data, which is present in the depolarization measurements to a larger extent than in the BR. Moreover, a critical aspect of this analysis is that aspherical particles passing through the OPC will scatter differently than a sphere of the same size, assumed in the OPC retrieval, and this causes an additional uncertainty on the particle true size. However it is likely that the lack of agreement has more profound reasons and that the assumptions made in the construction of our optical model are not fully reflected by reality. The most vulnerable among them is in our opinion the hypothesis of
having used, for each particle irrespective of the NAT or ICE type and, more critically, of its size, the same value for the polarized fraction X and, albeit not so critical for the determination of the depolarization, the depression factor C. In reality it



is known that, even for particles that share the same shape, the depolarization changes with their sizes, especially when they are comparable with the backscattered radiation. Furthermore, there is no reason to believe that solid particles share identical shapes. In contrast, laboratory studies (Grothe et al., 2006), have showed that synthesized aspherical NAT develops different morphologies depending on the growth conditions. The situation is even more complicated for ice particles, of which the extreme variety of shape is known, even in the case of pristine crystals whose morphology is influenced by the air temperature, atmospheric pressure, ventilation and ice supersaturation (Heymsfield et al., 2017).

It is well known that different shapes produce different polarization. This has been proven experimentally since the early works of Sassen and Hsueh (1998) and Freudenthaler et al. (1996) that showed how the lidar depolarization ratio in persisting contrails ranged from 10% to 70%, depending on the stage of their growth and on temperature. The dependence of the single particle depolarization on shape and size has also been studied theoretically by Liu and Mishchenko (2001) with the T-matrix approach. They showed not only that the depolarization value increases from 0 to a limiting value reached only when the particle size parameter $x=2\pi r/\lambda$ exceeds approximately 10 - which in our case implies that the particle size r should exceeds $1\mu$m - but also that this limiting value can vary from 15 to 70% depending on the assumptions made on the shape and aspect ratios of the particle.

The oversimplified assumption of a common X for all solid particles is the most likely reason why the optical model behaves so poorly in the simulation of depolarization, and should not be used for this case. This also leads us to revise the hypotheses made on the interpretation of the figure 2, admitting that the experimental points filling the triangle of vertices 0, $\delta_{TA}^{asph}$ and $\delta_{mol}$ represent PSC probably both in mixed phase or in solid phase where the particles have various different depolarizations. Thus $\delta_{TA}^{asph}$ should not be regarded as the limiting value, but the maximum of the limiting values for $\delta_{TA}$ as the clouds get thicker.

Nevertheless, the study of backscattering gives a good indication not only of the size transition at which cloud particles from STS may begin to be considered NATs, confirming the common wisdom on the microphysics of mixed NAT/STS clouds, but also - and this may be a more general result - suggesting what could be the range of Mie backscattering depression we should expect, on average, for aspherical particles of arbitrary shape. This is the positive result that can be gleaned from this study. At the same time it also suggests caution in the use of scattering models from aspheric particles in the calculation of the depolarization, when a single shape is used for them.

## 5 Conclusions

We have used an optical model to compute with Mie theory the backscatter and depolarization of mixed phase PSC. The model assumes that: i. PSC particles are solid above a threshold radius $R_{th}$; ii. Mie backscattering from solid particles is reduced by a depression factor C<1, which is common to all solid partiles irrespective of their size or shape or composition; iii. Only a polarized fraction X<1 of the backscattering from solid particles is polarized. This latter parameter has been estimated through the study of the statistics of PSC depolarization and backscatter ratio acquired over twelve years of lidar observations



in Antarctica. We have tested the model using a data set of coincident lidar, backscattersonde and OPC measurements from Antarctica and Arctic balloon flights.

The analysis suggested the optimal $R_{th}$ C and X parameters that best mach the observations. Unfortunately, while the agreement between modeled and measured backscatter is sufficiently satisfactory, it is poor for depolarization. Albeit it cannot be excluded a defective accuracy in depolarization data that prevent to fully demonstrate the validity of our assumptions, the most likely reason is an oversimplification in the hypothesis of common polarized fraction X for all the solid particles present in the size distribution. However, result for the backscatter coefficient simulation allows to constrain the model parameters $R_{th}$ and C into reasonable ranges and should be regarded as the positive result of the study.

The illustrated method can find application in other cases of externally mixed aerosols with coexistence of different phases, where the phases can be distinguished on the basis of the particle size.

*Code and data availability.* The Mcmurdo lidar data are available at the NDACC web site

ftp://ftp.cpc.ncep.noaa.gov/ndacc/station/mcmurdo/ames/lidar/.

The OPC data files and size distributions are reported at the web site hosting the Wyoming in situ data

http://www-das.uwyo.edu/~deshler/Data/Aer_Meas_Wy_read_me.htm

and can be downloaded from

ftp://cat.uwyo.edu/pub/permanent/balloon/Aerosol_InSitu_Meas/Ant_McMur.

The arctic balloonborne backscattersonde data are available from the author upon request. The software for Mie computation and data analysis is available from the author upon request.

*Author contributions.* FC was responsible for most of the writing, review and editing process, supported by all co-authors. FC and MS share the idea behind the article and the data analysis work. MB helped in software development. TD provided OPC data and PSD analysis. LDL and AS provided for the identification and quality check of the dataset.

*Competing interests.* No competing interests are present

*Acknowledgements.* The authors acknowledge the financial support by PNRA in the framework of the projects POAS (Particles and Ozone in the Stratosphere of Antarctica) and ACLIM (Antarctic Clouds Investigation by Multi-instrument measurements and modeling). The OPC measurements were supported by awards from the US National Science Foundation (NSF) which include OPP award numbers 9316774, 9615198, 9980594. Terry Deshler and Luca Di Liberto acknowledge a grant from the Short-Time-Mobility program of CNR, respectively in 2016 and 2009. The arctic balloon flights were supported by the Commission of the European Union through the Environment and Climate Program (contract ENV4-CT97-0523) and through the CIPA program (EVK2-CT-2000-00095).





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





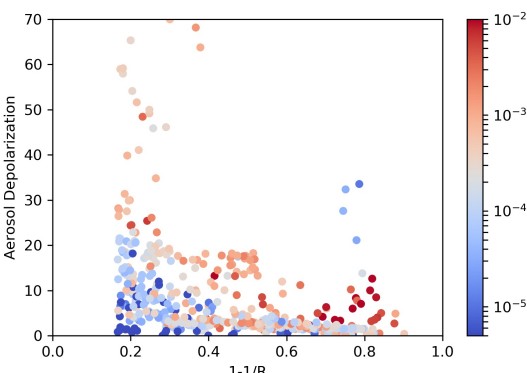

**Figure 1.** Scatterplot of aerosol depolarization vs 1-1/BR, where BR is the Backscatter Ratio. Data are from McMurdo lidar and Kiruna balloon flights backscattersonde, coincident with balloonborne OPC PSD measurements, fitted with mono or bimodal lognormals. The color codes the ratio of particles with radius larger than 1 $\mu$m to the total number of particles in the PSDs. We reports data points with BR greater than 1.2, $\beta_A^{cross}$ greater than $2\,10^{-6}km^{-1}sr^{-1}$ and temperature at the observation below 195 K.

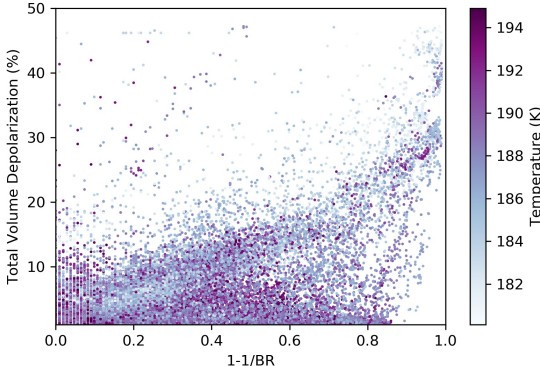

**Figure 2.** Scatterplot of Total Volume Depolarization $\delta_{TA}$ vs 1-1/BR, where BR is the Backscatter Ratio. Data are from McMurdo lidar and cover the winters from 1990 to 2002. The color codes the temperature of the observations.





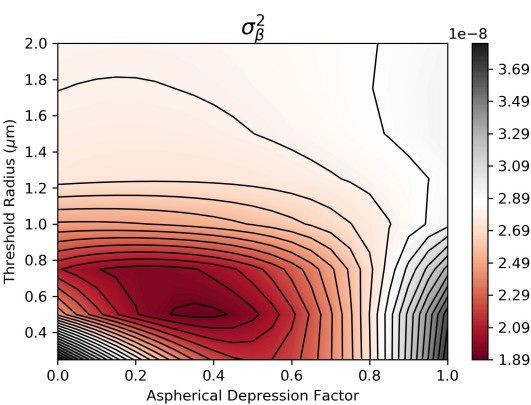

**Figure 3.** Contour plot of the MSE of the measured and modelled aerosol backscatter coefficient $\beta_A$.

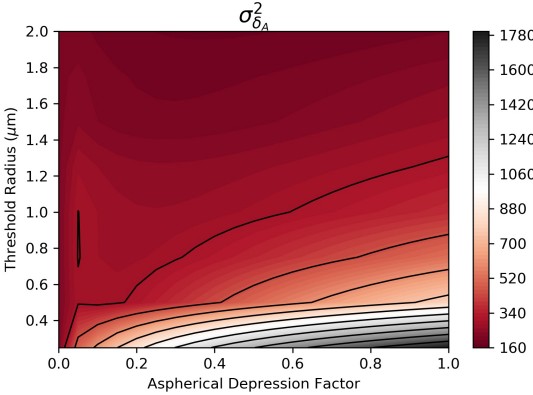

**Figure 4.** Contour plot of the MSE of the measured and modelled aerosol depolarization $\delta_A$.





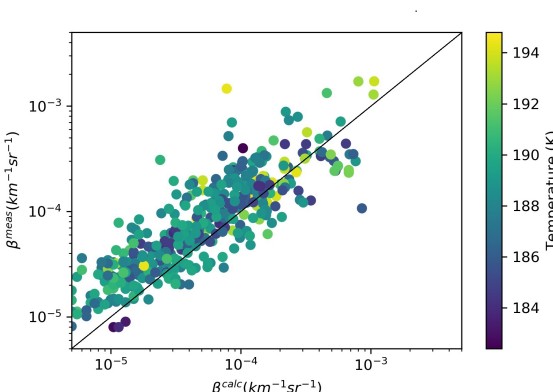

**Figure 5.** Scatterplot of computed vs measured particle backscatter coefficients $\beta_A$. The colour codes the temperature of the observations. We reports data points with BR greater than 1.2, $\beta_A^{cross}$ greater than $2\ 10^{-6} km^{-1} sr^{-1}$ and temperature at the observation below 195 K.

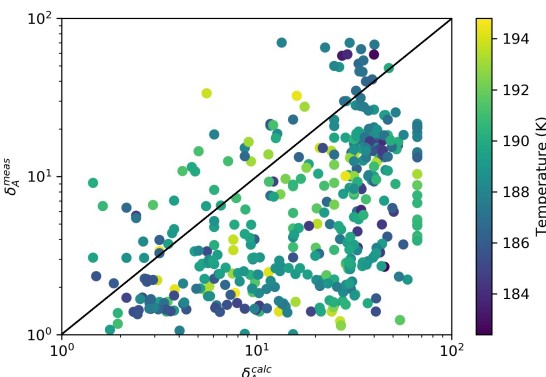

**Figure 6.** Scatterplot of computed vs measured particle backscatter coefficients $\delta_A$. The colour codes the temperature of the observations. We reports data points with BR greater than 1.2, $\beta_A^{cross}$ greater than $2\ 10^{-6} km^{-1} sr^{-1}$ and temperature at the observation below 195 K.