# Peer review of "A study of Mie scattering modelling for mixed phase Polar Stratospheric Clouds"

_Atmospheric Measurement Techniques, 2022_

## Author Comment (AC1)

**Response to: 'Comment on amt-2022-28', Anonymous Referee #1**

We thank the reviewer for her or his comments, and for the honest severity of the judgment which highlights shortcomings in our presentation that were not evident to us in the first draft. We will try to remedy this in the hope of a more benevolent reconsideration of the manuscript by the reviewer.

In fact, we would like to report under the consideration of the reviewer a result of the study that we think was not appreciated in full in the review and which we will therefore try to highlight below.

The reviewer states that "*Two empirical parameters are introduced to describe depolarization (X) and depression factor of solid particle backscattering (C).*" The parameters are actually three: 1. The threshold radius $R_{th}$; 2. the depression factor C; and 3. the depolarization fraction X. While the former two are identified by an error minimizing procedure, the third is retrieved by assuming a common particle depolarization $\delta_{TA}$ and estimating X by the procedure outlined in paragraph 2.3. It is the failure of that assumption on a common particle depolarization $\delta_{TA}$ which, in our opinion, jeopardizes the method presented and discussed in our paper as to lead to poor results in reproducing the depolarization. We think this is a result not properly acknowledged in the review.

We fully agree that "*the optical models lacks the basic physics consideration of light scattering.*" In fact, ours is an attempt, based on heuristic but reasonable considerations, to apply to the general case of particles of arbitrary shapes a method that gives exact results only in the case of spherical particles (or rather, of particles with an axis of symmetry along the direction of the incident radiation). We are comforted in this effort by the fact that Mie theory is still widely used – although improperly - for scattering modeling from aspheric particulate matter, probably also in view of the difficulty to find an appropriate shape representation for the aspherical particle habits.

We propose to change line 55-56 to:

"the use of Mie theory, notwithstanding its inability to reproduce depolarized backscattering, continues to be attractive and is used widely due to its convenient availability and simplicity (Yu et al., 2017;Yang et al., 2020; Dusing et al., 2021), particularly in processing of Optical Particle Counter and Sizers (Keener et al., 2007, Chalut et al., 2008; Mahrt et al., 2019), to treat non-spherical particles (especially those in random orientation), as if they were spheres to which, strictly, Mie theory is only applicable (Borrmann et al., 2000)."

We understand the encouragement of the reviewer "*to examine their work under the light of assuming true nonspherical particle contribution*". In fact this has been done in the past (Scarchilli et al., 2005) by some of the same authors of this paper, on a subset of the dataset here presented, but with the present work our aim was not "*to gain any knowledge about the microphysical and optical properties of PSC*". We rather tempted to define the limits of applicability of a – we grant – heuristic model.

Finally, we must face what is in our opinion the most severe criticism of the reviewer: the accusation of "*trying to fit the data with some arbitrary empirical parameters*". On the contrary, we think that the parameters chosen in our heuristic model are not chosen arbitrarily but rather have firm physical justifications:

1. Threshold Radius $R_{th}$; It is known since long that solid NAT particles have sizes several times larger than liquid STS particles (Voigt et al., 2000; Zhu et al, 2017) so the choice of a threshold radius below which particles are considered liquid, and above which are solid seemed to us a natural, "physical" one. The minimizing procedure for the backscattering retrieval identifies $R_{th}$ in the range expected and reported by studies in the literature.

2. Depression factor C; Theoretical computations have largely shown that backscattering peak for aspherical, randomly oriented particles, is substantially depressed with respect to their Mie spherical counterpart. In general, aspherical particles tend to exhibit enhanced scattering at intermediate scattering angles and reduced backscattering, whereas phase function differences at forward-scattering angles are often negligibly small (Mishchenko et al., 2000). Again our choice is based on physical results. The minimizing procedure for the backscattering retrieval identifies C in the range expected and reported by studies in the literature (e.g. Mishchenko, 2009).

3. depolarization fraction X; The assumption of a constant, common particle depolarization $\delta_{TA}$ for relies on the fact that, for a specific shape, and above few size parameters, the $\delta_{TA}$ is largely independent on particle dimension (e.g. Liu et al, 2001). Hence we deemed reasonable to assume that the procedure outlined in paragraph 2.3. could result in the identification of this parameter. We want to highlight the fact that this result is independent of the use of Mie theory. It is based solely on the assumption that if aspheric particles share an identical morphology then they share the same $\delta_{TA}$ and that all the aspherical particles have an identical morphology. In our opinion, it is the violation of this assumption that leads to the failure of the method to reproduce the observed depolarization. We believe this is the significant result of our study. We point out that, if confirmed, it would prevent the application of aspheric particle diffusion computation codes based on hypothesis on identical particle morphology.

We have thus tried to better justify the choice of parameterization of our model by adding the following to line 175:

"In conclusion, although Mie theory lacks the capability of accounting for the two main causes of lidar depolarizations, i.e particle asphericity and multiple scattering, this latter can be neglected on the basis of the relatively low particle density in PSC. Concerning the former, our heuristic model try to mimic the particle asphericity effect by i. depressing the Mie result by the factor C<1, as nonspherical particles tend to exhibit reduced backscattering with respect to their spherical couterparts (Mishchenko et al., 2000); ii. Assuming that aspherical particles of a specific shape attains constant $\delta_{TA}$, independent on particle dimension, as it has been demonstrated to be the case above few size parameters (Liu et al, 2001). Hence we deem reasonable to assume that the procedure outlined in the following paragraph 2.3. could result in the identification of such value."

We trust that these responses may induce the reviewer to change his or her opinion on our work.

Bibliography:

Borrmann, S., Luo, B., and Mishchenko, M.: Aplication of the T-matrix method to the measurement of aspherical (ellipsoidal) particles with forward scattering optical particle counters, J. Aerosol Sci., 31, 789–799, 2000.

Chalut KJ, Giacomelli MG, Wax A. Application of Mie theory to assess structure of spheroidal scattering in backscattering geometries. J Opt Soc Am A Opt Image Sci Vis. 2008 Aug;25(8):1866-74. doi: 10.1364/josaa.25.001866. PMID: 18677348; PMCID: PMC2840708.

Düsing, S., Ansmann, A., Baars, H., Corbin, J. C., Denjean, C., Gysel-Beer, M., Müller, T., Poulain, L., Siebert, H., Spindler, G., Tuch, T., Wehner, B., and Wiedensohler, A.: Measurement report: Comparison of airborne, in situ measured, lidar-based, and modeled aerosol optical properties in the central European background – identifying sources of deviations, Atmos. Chem. Phys., 21, 16745–16773, https://doi.org/10.5194/acp-21-16745-2021, 2021.

Keener JD, Chalut KJ, Pyhtila JW, Wax A. Application of Mie theory to determine the structure of spheroidal scatterers in biological materials. Opt Lett. 2007 May 15;32(10):1326-8. doi: 10.1364/ol.32.001326. PMID: 17440576.

Liu, L. and Mishchenko, M. I.: Constraints on PSC particle microphysics derived from lidar observations, Journal of Quantitative Spectroscopy and Radiative Transfer, 70, 817–831, 2001.

Mahrt, F., Wieder, J., Dietlicher, R., Smith, H. R., Stopford, C., and Kanji, Z. A.: A high-speed particle phase discriminator (PPD-HS) for the classification of airborne particles, as tested in a continuous flow diffusion chamber, Atmos. Meas. Tech., 12, 3183–3208, https://doi.org/10.5194/amt-12-3183-2019, 2019.

Mishchenko, M.I., J.W. Hovenier, and L.D. Travis (Eds.), 2000: Light Scattering by Nonspherical Particles: Theory, Measurements, and Applications. Academic Press.

Mishchenko, M. I.: Electromagnetic scattering by nonspherical particles: A tutorial review, Journal of Quantitative Spectroscopy and Radiative Transfer, 110, 808–832, https://doi.org/https://doi.org/10.1016/j.jqsrt.2008.12.005, light Scattering: Mie and More Commemorating 100 years of Mie's 1908 publication, 2009.

Scarchilli Claudio, Alberto Adriani, Francesco Cairo, Guido Di Donfrancesco, Carlo Buontempo, Marcel Snels, Maria Luisa Moriconi, Terry Deshler, Niels Larsen, Beiping Luo, Konrad Mauersberger, Joelle Ovarlez, Jim Rosen, and Jochen Schreiner, "Determination of polar stratospheric cloud particle refractive indices by use of in situ optical measurements and T-matrix calculations," Appl. Opt. 44, 3302-3311 (2005)

Voigt, C., et al., 2000. Nitric acid trihydrate (NAT) in polar stratospheric clouds, Science, 290, 1756.

Yang Haifeng and Zhi-Yun Li The Effects of Dust Optical Properties on the Scattering-induced Disk Polarization by Millimeter-sized Grains 2020 Astrophysical Journal 889 15.

Yu X, Shi Y, Wang T, Sun X (2017) Dust-concentration measurement based on Mie scattering of a laser beam. PLoS ONE 12(8): e0181575. https://doi.org/10.1371/journal.pone.0181575

Zhu, Y., Toon, O. B., Lambert, A.,Kinnison, D. E., Bardeen, C., & Pitts, M. C.(2017). Development of a polar stratospheric cloud model within the Community Earth System Model: Assessment of 2010 Antarctic winter.Journal of Geophysical Research:Atmospheres, 122. https://doi.org/10.1002/2017JD027003